Physique and performance in male sitting volleyball players: implications for classification and training

Cavedon Valentina valentina.cavedon@univr.it 1
Brugnoli Chiara 1
Sandri Marco 1
Bertinato Luciano 1
Giacobbi Lorenzo 2
Bolčević Filip 3
Zancanaro Carlo 1
Milanese Chiara 1
1 Department of Neurosciences, Biomedicine and Movement Sciences, University of Verona , Verona , Italy
2 Marche Regional Committee, Italian Paralympic Committee , Ancona , Italy
3 Faculty of Kinesiology, University of Zagreb , Zagreb , Croatia
Nuhmani Shibili
Electronic publication date: 2022 Oct 7
Publication date: 2022
Volume: 10
Electronic Location ID: e14013
Received 2022 Jun 20; Accepted 2022 Aug 15
Copyright: ©2022 Cavedon et al.
Copyright year: 2022
Copyright holder: Cavedon et al.
License: This is an open access article distributed under the terms of the Creative Commons Attribution License, which permits unrestricted use, distribution, reproduction and adaptation in any medium and for any purpose provided that it is properly attributed. For attribution, the original author(s), title, publication source (PeerJ) and either DOI or URL of the article must be cited.
License URL: https://creativecommons.org/licenses/by/4.0/

Keywords: Paralympic athletes, DXA, Anthropometry, Sport-specific field tests, Athletes with a lower limb amputation, Adapted sports

Funding: The authors received no funding for this work.

==============================
Background

This study assessed whether anthropometry, physical fitness and sport-specific sprint performance vary across the three groups of sitting volleyball (SV) athletes (athletes with a disability (VS1), athletes with a minimal disability (VS2) and able-bodied SV athletes (AB)) in order to explore the validity of the current system of classification. This study also investigated how the anthropometric and physical fitness characteristics of athletes relate to their sprint performance.

Methods

Thirty-five SV male athletes aged 37.4 ± 10.8 years and practicing SV at a national/international level volunteered for this study. Testing consisted in the evaluation of linear anthropometry, physical fitness (body composition by-means of dual-energy X-ray absorptiometry and upper-body strength) and sprint performance (5-meter sprint tests, agility test and speed and endurance test).

Results

Athletes in the three groups differed in fat mass percentage (%FM) which was higher in VS1 versus AB at the sub-total level (+9%), in the arms (+15%) and in the non-impaired leg (+8%) regions. Greater hand span, greater length of the impaired lower leg, lower %FM at both the sub-total and regional level and a higher level of strength in the upper body are all associated with better performances in the considered sprint tests (P < 0.05 for all). These results do not confirm the validity of the current system of classification of athletes adopted in SV. Professionals dealing with SV athletes should include specific exercises aimed at improving whole-body and regional body composition and the strength of the trunk and upper limbs in their training programs.

Introduction

Sitting volleyball (SV) is an intriguing example of a popular, accessible and inclusive Paralympic team sport. SV retains most of the major rules and scoring of standing volleyball, but introduces some adaptations in consideration of the presence of athletes with a physical impairment. These adaptations involve playing by sitting directly on the court (10 × 6 m), which is smaller than the standing volleyball court and with a lower net (1.15 m for men and 1.05 m for women) (World ParaVolley, 2022). Other adaptations are that athletes cannot lift their buttocks from the floor while bouncing the ball and that they have the possibility to attack or block serves (World ParaVolley, 2022).

As in other Paralympic sports, a classification system is also adopted in SV in order to ensure fair and balanced competitions between athletes with different types and severity of physical impairments. In SV, athletes who are eligible to compete are classified into two functional classes according to the severity of their impairment: the volleyball sitting 1 (VS1 Class; prior to January 2018 this functional class was referred as “Disability Class”) and the volleyball sitting 2 (VS2 Class; prior to January 2018 this functional class was referred as “Minimal Disability Class”) (World ParaVolley, 2018). The VS1 Class includes athletes who have an impairment that more significantly affects the core functions in SV (e.g., athletes with a lower limb amputation at the transfemoral or transtibial level), while the VS2 Class is reserved for athletes whose impairment minimally affects the core functions in SV (e.g., athletes with a partial foot amputation) (World ParaVolley, 2018). According to the SV rules, during the international competitions at least one player per team in the VS2 Class must be on court during play (World ParaVolley, 2022). Furthermore, in order to increase general participation in SV and to create more competitive opportunities for SV athletes, able-bodied people are also permitted to compete in SV in national competitions, thereby bringing athletes together without anyone’s abilities limiting their opportunity and promoting integration.

SV is a fast and dynamic game in which athletes are required to move quickly on the playing court using the hands in order to get into position early enough to play effectively (Marszalek et al., 2015; Jeoung, 2017; Yüksel & Sevindi, 2018). As in other team sports, when moving on the court SV players have to accelerate, decelerate, and change direction throughout the game in response to a stimulus like the movement of the ball and an opposing player’s movement (Sheppard & Young, 2006). Accordingly, the sport-specific sprint performance in SV requires athletes to be exceptional movers in forward, backward, lateral and multidirectional movements. Moreover, the dynamicity of the play requires them to have high physical fitness levels (i.e., high upper body strength and power, speed, agility and good balance in the sitting position) (Ko & Kim, 2005; Lee & Kim, 2010; Marszalek et al., 2015; Molik et al., 2017; Jeoung, 2017; Yüksel & Sevindi, 2018). Given their accessibility, ease of use and ability to mimic the competition-relevant SV skill tasks, field tests are commonly used by sport scientists (Molik et al., 2013; Marszalek et al., 2015; Jeoung, 2017; Ahmadi, Uchida & Gutierrez, 2019; Ahmadi, Gutierrez & Uchida, 2020b; Ahmadi et al., 2020) to evaluate physical fitness levels and the sport-specific sprint performance.

In spite of the sport’s apparent popularity compared with other Paralympic team sports, today there are relatively few publications focusing on SV athletes (Marszalek et al., 2015; Ahmadi, Uchida & Gutierrez, 2019; Petrigna et al., 2022). In fact, through a literature research carried out in January 2021 and using the keywords “sitting” AND “volleyball” in the electronic database PubMed, only 12 articles were found, while the keywords “wheelchair” AND “basketball” or “wheelchair” AND “rugby” or “wheelchair” AND “tennis” yielded 247, 123 and 76 papers, respectively. Of the available SV literature, most scientific data focused on injury prevention (Macedo et al., 2019; Ahmadi, Gutierrez & Uchida, 2020a; Krzysztofik et al., 2021; Jarraya et al., 2021; Gaweł& Zwierzchowska, 2021; Zwierzchowska et al., 2022a; Zwierzchowska et al., 2022b) or on the implications of anthropometry, physical fitness, sport-specific sprint performance and game efficiency on the classification and training of athletes (Marszalek et al., 2015; Molik et al., 2017; Jeoung, 2017; Ahmadi et al., 2020). However, to date there are still several aspects of anthropometry and physical fitness, as well as their possible implications on classification and training, that have not yet been investigated. For example, to date in literature there is no information about the importance of the length of some crucial body segments of SV athletes (e.g., the hand and the lower limb). It is reasonable to assume that the length of some body segments (the hand and the lower leg) would have an impact on sprint performance as SV athletes exert strength through contact of their hands and feet/foot on the court. This could be of special interest in terms of classification and training because it is important to bear in mind that SV is also open to athletes with upper limb impairments and that all athletes with a lower limb amputation are assigned the same functional class (i.e., the VS1 Class) regardless of the level of amputation.

Body composition is a crucial component of physical fitness and its relevance to performance in sport has long been appreciated with special attention given to the total and regional proportion of body fat mass (%FM) (Ackland et al., 2012). However, as of yet aspects related to body composition in SV athletes have not received much attention from the scientific community. Based on the literature dealing with able-bodied athletes (Ackland et al., 2012), we assume that SV players would benefit from low %FM to move their bodies on the court.

When training athletes with different types of disability and different degrees of severity, physical conditioners and coaches would benefit from scientific data on the physical fitness characteristics of athletes, according to their functional class and type of impairment, in order to make the most effective decisions. A common limitation of several scientific papers dealing with Paralympic sports, including SV (Jadczak et al., 2010; Marszalek et al., 2015; Jeoung, 2017), is that authors did not report information about the type of impairment of athletes in their study groups. So, in many cases, it is not known whether the results refer to athletes with an impairment in one lower limb (e.g., in the case of athletes with unilateral lower limb amputation), in both lower limbs (e.g., in the case of athletes with bilateral lower limb amputation or spina bifida), and if the upper limbs are impaired. A second limitation of some papers in this research area is that the study groups are not homogeneous in terms of the type of disability of the athletes and have included athletes of both genders, thereby increasing variability.

In order to fill some of these important gaps in the scientific literature, we assessed the physique and performance of 35 SV male athletes by measuring the length of a number of body segments including the hand and the lower limb, a number of physical fitness characteristics (i.e., body composition, assessed by-means of dual-energy X-ray absorptiometry (DXA) and upper body strength) and the sport-specific sprint performance. In this study we tried to reduce variability as much as possible by including only athletes with one lower limb impairment; more specifically from the VS1 functional class we only included athletes with unilateral lower limb amputation, which is a typical disability of SV players.

The aim of this study was therefore twofold. First, to assess whether anthropometry, physical fitness and sport specific sprint performance vary across the three groups of functional classes admitted in SV competitions (i.e., VS1 functional class, VS2 functional class and able-bodied SV athletes) in order to explore the validity of the current system of classification. Second, to investigate how the anthropometric and physical fitness characteristics of athletes relate to their sport-specific sprint performance.

Materials & Methods

Participants

Considering the prospective cross-over design of this study, the sample size was estimated a priori using the formula of the F test of the one-way analysis of variance (ANOVA). The minimum sample size was calculated considering a primary outcome of DXA-measured body composition (i.e., the %FM at the whole-body level) for a population of male athletes with unilateral lower limb amputation. In the study of Cavedon et al. (2021) the mean value of the whole-body %FM was 22% with a standard deviation of 4%. We considered a minimal relevant absolute difference of 5% corresponding to an eta-squared effect size (η2) of 0.23 (i.e., a medium effect size according to Cohen) (Cohen, 1988). Setting the type-I error to α = 0.05, the effect size to η2 = 0.23 and the power to 80% (i.e.,  β = 0.20), the minimum required total sample size was 36 subjects. The sample size calculation was performed using G ∗Power ver. 3.1.9.6 (Faul et al., 2009).

Inclusion criteria was male gender, age 18 or over and practicing SV at a competitive level for at least 6 months prior to testing. For athletes with a physical impairment, further inclusion criteria were: the diagnosis of a lower limb amputation allowing them to compete in the VS1 functional class or the diagnosis of a lower limb physical impairment allowing them to compete in the VS2 functional class.

Sitting volleyball athletes were recruited by contacting directly all the national and international teams closest to the test site (up to 800 km away) and we were able to recruit a random sample of thirty-five SV male athletes aged 37.4  ± 10.8 years. For the purposes of this study, athletes were divided into three groups: VS1 functional class (n = 17; age = 36.3  ± 11.3 years; amount of training = 4.4  ± 1.5 h per week), VS2 functional class (n = 9; age = 40.0  ± 9.1 years; amount of training = 4.4  ± 1.8 h per week) and able-bodied SV athletes (n = 9; age = 36.8  ± 12.2 years; amount of training = 3.6  ± 1.4 h per week). Eleven athletes were members of the Italian SV National Team and seven athletes were members of the Croatian SV National Team. All athletes were practicing SV at a national/international level and all of them were actively training (estimated mean training time per week, 4.2  ± 1.5 h).

In the group of athletes with a physical impairment, the origin of disability was acquired (n = 22) or congenital (n = 4). In the case of acquired disability, the average duration of injury was 12.1  ± 13.9 years. Disabilities were comprised of unilateral transfemoral amputation (n = 5), unilateral transtibial amputation (n = 12), amputation of the foot (n = 2) and other types of minimal disability which meet the inclusion criteria for sport class VS2 (n = 7).

The study was conducted in accordance with the Declaration of Helsinki, and the protocol was approved by the Institutional Review Board of the University of Verona (Protocol number: 18198, 05/04/2013). All participants were volunteers and signed an informed consent form.

Testing procedures

Testing took place on the same day and on the same University structure, in the late morning/early afternoon, after a 3-4 h fast. All participants were asked not to undertake any strenuous physical activity the day before each measurement session, and they were also required not to undertake any exercising on the day of the measurements.

The experimental protocol consisted in the following standardized order; collecting the athletes’ general information through a face-to-face questionnaire; the assessment of anthropometry; physical fitness and sport-specific sprint performance.

Face-to-face questionnaire

All athletes completed a face-to-face questionnaire to confirm the participants’ eligibility criteria and to collect information about demographics, type and severity of impairment, origin of disability (congenital or acquired), duration of injury (in the case of acquired disability), sport classification, years of experience in sitting volleyball and weekly amount of training expressed in hours.

Anthropometric assessment

Anthropometric data were taken by one operator using conventional criteria and measuring procedures (Lohman, Roche & Martorell, 1988). All anthropometric measurements were collected according to conventional criteria and measuring procedures (Lohman, Roche & Martorell, 1988). In order to adopt an ecological approach and according to previous literature in this field (Cavedon, Zancanaro & Milanese, 2015; Cavedon, Zancanaro & Milanese, 2018), the athletes were measured while sitting on the floor with their lower limb extended, assuming this is more representative of the real situation during play. For the sitting height, two measurements were taken with a Harpenden anthropometer (Holtain Ltd., Crymych, Pembrokeshire, UK): (1) the sitting height (SITH1; Fig. 1A), measured as the vertical distance from the vertex of the head to the floor; (2) the vertical grip reach from a seated position (SITH2; Fig. 1B), measured as the maximal distance from the tip of the dactylion III to the floor, with the upper arms extended overhead as much as possible.

Figure 1 Procedures of anthropometric measurements.

(A) SITH1 the vertical distance from the vertex of the head to floor; SE, shoulder-elbow length; EA, elbow-hand length. (B) SITH2 the vertical grip reach from a seated position, was measured as the maximal distance from the tip of the dactylion III at the maximum to the floor, with the upper arms extended overhead as much as possible. (C) HS, hand span. (D) AS, arm span; (E) NI_L, non-impaired leg length; (F) I_L, impaired leg length. (G) HR, head region; AR, arms region; TR, trunk region; NI_LR, non-impaired leg region; I_LR, impaired leg region.

The following body dimensions were measured with a Harpenden anthropometer (Holtain Ltd., Crymych, Pembrokeshire, UK) to the nearest 0.1 cm: arm span, shoulder-elbow length, elbow-hand length, hand span, leg length (Figs. 1A, 1C–1E). For athletes with unilateral lower limb amputation (n = 17), the impaired leg length was also measured as the distance from the buttocks to the end of the stump (Fig. 1F).

Physical fitness assessment

The physical fitness assessment consisted in the evaluation of body composition and upper body strength.

Body composition was measured using DXA using a total body scanner (QDR Horizon, Hologic MA, USA; fan-beam technology, Hologic APEX software version 5.6.1.2). In our laboratory quality control of the DXA scanner is performed at least once weekly and always before actual use by means of an encapsulated spine phantom (Hologic Inc., Bedford, MA, United States) to document the stability of the DXA performance (Lewiecki et al., 2004). Athletes undertook total-body DXA scanning according to “The Best Practice Protocol for the assessment of whole-body body composition by DXA” (Nana et al., 2015). All DXA scans were carried out and analysed by the same trained research technician to ensure consistency as described elsewhere (Cavedon, Zancanaro & Milanese, 2020; Cavedon et al., 2021). The percentage of fat mass (%FM) assessed at the sub-total level (whole-body less head) and in the arms, trunk, non-impaired leg and impaired leg regions were considered for analysis (Fig. 1G).

The upper body strength was evaluated through a battery of four field tests (strength field tests: sit-ups test, modified plank test, seated chest pass test and handgrip strength test).

Prior to testing an operator gave detailed instructions and an adequate technique demonstration of each test. After which, field tests were performed following a standardized 15-minutes warm-up consisting in low to medium intensity sport-specific sprints, acceleration and agility drills as well as mobility and stretching exercises involving the major muscle groups.

All test trials were completed at the same indoor gym with a complete rest between each test. The temperature at the test place was kept constant throughout the duration of the tests. During the tests athletes wore the same sport clothes.

An explanation of the experimental set-up and testing procedure of each test is provided below.

Strength field tests

Sit-ups test

According to Yüksel & Sevindi (2018), athletes lay on their backs on the mat with their knees bent, the soles of the feet flat on the mat, the hands positioned on each side of the hips, and the fingers fully extended on the mat. The legs (or the residual leg in the case of athletes with lower limb amputation) were supported by an operator as to keep the knees bent. The athletes were asked to rise until the scapula bottom level is detached from the floor, and do as many sit-ups as they could in 30 s.

Modified plank test

Each athlete was then asked to assume the plank position with elbows in contact with the ground and the humerus forming a perpendicular line to the horizontal plane, the forearms in neutral position and the hands directly in front of the elbows. In the plank position athletes assumed a rigid anatomical body position allowing only their forearms and toes to support their body. The test was performed once and consisted in holding the plank position as long as possible (Strand et al., 2014). During the test, verbal cues were provided to the athletes in order to promote form adherence for test validity.

Seated chest pass test

The seated chest pass test was used to assess the power of the upper body of SV players according to the literature (Molik et al., 2013; Marszalek et al., 2015; Jeoung, 2017; Ahmadi, Uchida & Gutierrez, 2019). The athletes sat on the floor with their back against a wall, the legs in an extended position and the feet 60 cm apart. Athletes were asked to hold a 4 kg medicine ball with both hands in front of the chest and with their forearms parallel to the ground. Athletes were then asked to throw the medicine ball straight forward as strongly and as far as they could while maintaining their back part touching the wall. The test consisted of two trials with a 45–60 s rest between them (Stockbrugger & Haennel, 2001) and the best distance thrown was recorded.

Handgrip strength test

Prior to conducting the test, athletes performed three preliminary trials at very low intensity in order to avoid muscle fatigue. All measurements were performed in a seated position, using a portable hydraulic dynamometer (SAEHAN, Chinesport Spa, Udine, Italia), as previously described by Ahmadi, Uchida & Gutierrez (2019); Ahmadi, Gutierrez & Uchida (2020b). Athletes were placed in a seated position with the elbow bent (90°) and in touch with the trunk. The test consisted in gripping the dynamometer as hard as possible until the operator gave a vocal stop signal. Athletes performed the handgrip strength test with both hands (3 trials each) with a 2–5 s rest between each trial (Jeoung, 2017). For each hand the best trial was recorded and expressed in kilograms.

Sport-specific sprint performance

The sport-specific sprint performance was evaluated through a battery of four field tests (5m forward sprint test, 5m backward sprint test, modified agility T-test and speed and endurance test) assessing speed, agility and endurance in the sport-specific sprint abilities.

5m forward sprint test and 5m backward sprint test

For the 5m forward sprint test athletes started from a stationary position and moved in a forward direction for a distance of 5 m as quickly as possible according to Marszalek et al. (2015) (Fig. 2A). Similarly, for the 5m backward sprint test the athletes moved in a backward direction for the 5 meters’ distance as quickly as possible (Fig. 2A). Both the 5m forward sprint test and the 5m backward sprint test consisted of two trials.

Figure 2 Experimental set-up of the sport-specific field tests.

(A) Experimental set-up for the 5m forward sprint test and for the 5m backward sprint test. (B) Experimental set-up for the modified agility T-test. (C) Experimental set-up for the speed and endurance test.

Modified agility T-test

The modified agility T-test consisted of two trials and was conducted based on the protocol outlined by Sassi et al. (2009). For the modified agility T-test (Fig. 2B), athletes were seated behind the start line A and moved forward to cone B touching the base of the cone with their right hand; then they shuffled to the left to cone C touching its base with the left hand; after that, athletes shuffled to the right to cone D touching the base with the right hand; then, athletes shuffled back to the left to cone B touching the cone base; finally, athletes moved backward as fast as possible to return back to line A.

Speed and endurance test

The speed and endurance test consisted of two trials and was employed to assess the endurance and speed abilities of the athletes. According to Marszalek et al. (2015), athletes began from the seated position behind the start at cone A; afterwards, each athlete shuffled, as quickly as possible, back and forth between cone A and cones B, C, D, E, F, and G, respectively (Fig. 2C). During the test, athletes were required to touch the base of all the cones.

The time to complete all sprint tests was assessed through tripod-mounted photocells (Polifemo Light Radio, Microgate SRL, Bolzano, Italy) and for each test the best time was recorded.

Statistical analysis

Normality of data was assessed using the Shapiro–Wilk test and descriptive statistics (mean ± standard deviation) were computed for all variables using standard procedures.

The ANOVA followed by the post-hoc test with Bonferroni’s correction for multiple comparisons was used to assess the differences between groups (i.e., VS1 functional class, VS2 functional class and able-bodied SV athletes). The Levene’s Test of Equality was applied to check homogeneity of variance between groups. The ratio of variance explained in the dependent variable by predictor while controlling for other predictors (eta squared, η2) was used to calculate the effect size in the ANOVA and the effect size values were interpreted as small (η2 = 0.02), medium (η2 = 0.13), and large (η2 = 0.26) according to Cohen’s guidelines (Cohen, 1988).

In the whole sample, the degree of association between two continuous variables, accounting for the effect of the assigned group, was measured by partial correlation (rPC). Furthermore, in the sub-group of athletes with unilateral lower limb amputation (n = 17), the Pearson’s product-moment correlation coefficient (r) was used to assess the relationship between both the impaired leg length and the %FM in the impaired leg and the sport-specific sprint performance.

All analysis was performed with SPSS v. 26.0 (IBM Corp., Armonk, NY, USA) and the statistical significance was set at P ≤ 0.05.

Results

Descriptive statistics (mean ± standard deviation) relative to the anthropometric, body composition and performance results obtained in the aggregate sample as well as in the three groups (i.e., VS1 athletes, VS2 athletes and able-bodied SV athletes) are reported in Table 1.

Table 1 Anthropometric, body composition and performance variables assessed in the aggregate sample and in the three functional groups. Statistically significant P-values are in bold. Data are means ± standard deviation.

	Aggregate sample (n = 35)		VS1 Class (n = 17)		VS2 Class (n = 9)		AB SV athletes (n = 9)		One-way ANOVA			
	Mean	SD	Mean	SD	Mean	SD	Mean	SD	F	P	η 2	
Anthropometry												
SITH1 (cm)	93.6	3.9	94.2	3.9	93.9	4.6	92.1	3.0	0.880	0.425	0.052	
SITH2 (cm)	144.6	8.6	144.8	7.7	147.1	12.9	141.8	4.0	0.856	0.434	0.051	
Arm span (cm)	187.7	9.0	188.3	9.7	187.1	9.5	187.2	8.2	0.064	0.938	0.004	
Shoulder-elbow length (cm)	38.3	2.4	38.8	2.6	38.2	2.6	37.5	1.8	0.814	0.452	0.048	
Elbow-hand lenght (cm)	49.2	2.2	49.6	2.3	48.9	2.4	48.7	2.0	0.488	0.618	0.030	
Hand span (cm)	45.5	5.7	45.7	5.9	46.4	5.4	44.2	6.1	0.346	0.710	0.021	
Non-impiared leg length (cm)	108.4	6.3	108.8	7.6	108.6	5.7	107.2	4.7	0.208	0.813	0.013	
Body composiiton												
Sub-total %FM	25.0	7.6	27.9	5.8	25.1	7.6	19.4	8.1	4.502	0.019	0.220	
Arms %FM	27.2	9.3	32.8	6.4	25.8	8.4	18.0	7.0	12.937	<0.001	0.447	
Trunk %FM	26.0	8.5	28.6	7.4	26.2	8.5	20.7	9.0	2.822	0.074	0.150	
Non-impaired leg %FM	23.1	6.8	25.9	3.7	23.2	8.2	17.8	7.4	5.110	0.012	0.242	
Strenght Tests												
Sit-Ups Test (n)	39.8	9.2	36.2	8.2	38.5	5.1	47.8	9.3	6.376	0.005	0.291	
Modified Plank Test (s)	103.9	47.3	84.3	42.9	116.0	50.0	130.3	40.3	3.607	0.039	0.189	
Seated Chest Pass Test (m)	5.1	0.7	5.2	0.8	4.9	0.8	4.9	0.4	0.766	0.473	0.046	
HST_Dominant hand (kg)	46.2	9.4	47.3	9.8	45.8	9.0	44.6	9.9	0.248	0.782	0.015	
HST_Non-dominant hand (kg)	45.4	9.7	46.9	10.9	45.0	10.2	43.0	6.8	0.485	0.620	0.029	
Sport-Specific Field Tests												
5m Forward Sprint Test (s)	2.7	0.5	2.6	0.5	2.8	0.4	2.8	0.4	0.625	0.542	0.038	
5m Backward Sprint Test (s)	2.5	0.5	2.7	0.6	2.4	0.3	2.2	0.5	2.870	0.071	0.152	
Modified Agility T-test (s)	12.0	2.0	12.1	1.8	12.4	2.2	11.2	2.0	0.846	0.439	0.050	
Speed and Endurance Test (s)	28.7	5.6	28.9	5.4	28.5	6.4	28.3	5.8	0.041	0.960	0.003	
Notes.

VS1 Class class which includes athletes with an impairment that more significantly affects the core functions in sitting volley

VS2 Class class which includes athletes with an impairment that more significantly affects the core functions in sitting volley

AB able-bodied

SV sitting volleyball athletes

ANOVA Analysis of Variance

SD Standard Deviation

F F-value

P P value

η2 eta squared

SITH1 the vertical distance from the vertex of the head to floor

SitH2 the vertical grip reach from a seated position

%FM DXA-measured percentage of fat mass

HST Handgrip Strength Test

Difference in physique and performance across the three groups

The one-way ANOVA showed no statistically significant differences between the three groups in age (F = 0.349, P = 0.708; η2 = 0.02) and weekly amount of training (F = 1.076, P = 0.353; η2 = 0.06). Similarly, the three groups were similar for all the considered anthropometric variables (P > 0.05 for all; Table 2).

Table 2 Partial correlation coefficients (rPC) between general characteristics, anthropometry, physical fitness and sport-specific sprint performance calculated on the whole sample (n = 35). Statistically significant correlations are in bold.

	5 m forward sprint test	5 m backward sprint test	Modified agility T-test	Speed and endurance test	
General characteristics					
Age	0.121	0.251	0.171	−0.024	
SV experience	−0.280	−0.110	−0.189	−0.266	
Amount of training	−0.117	−0.044	−0.033	−0.073	
Anthropometry					
SITH1	0.118	0.021	0.027	−0.168	
SITH2	0.029	−0.081	−0.100	−0.236	
Arm span	−0.008	−0.093	−0.162	−0.275	
Shoulder-elbow length	0.063	−0.120	−0.130	−0.301	
Elbow-hand length	0.113	0.003	−0.046	−0.216	
Hand span	−0.240	−0.381*	−0.367*	−0.378*	
Non-impaired leg	0.295	0.110	0.107	0.012	
Body composition					
Sub-total %FM	0.345*	0.424*	0.471**	0.483**	
Arms %FM	0.253	0.341*	0.431*	0.417*	
Trunk %FM	0.354*	0.456**	0.455**	0.485**	
Non-impaired %FM	0.265	0.262	0.432*	0.398*	
Strength Field Tests					
Sit-Ups Test	−0.375*	−0.312	−0.483**	−0.352*	
Modified Plank Test	−0.358*	−0.350*	−0.334	−0.252	
Seated Chest Pass Test	−0.134	−0.270	−0.403*	−0.414*	
HDG_Dominant hand	−0.188	−0.264	−0.339*	−0.401*	
HDG_Non-dominand hand	−0.139	−0.263	−0.357*	−0.365*	
Notes.

SV Sitting Volley

SITH1 the vertical distance from the vertex of the head to floor

SITH2 the vertical grip reach from a seated position

%FM DXA-measured percentage of fat mass

HDG Handgrip Strength Test

* P < 0.05.

** P < 0.01.

*** P < 0.001.

As reported in Table 1, the one-way ANOVA revealed statistically significant differences in the %FM assessed at the sub-total level as well as in the arms and in the non-impaired leg regions (P < 0.05 for all). The post hoc analysis with Bonferroni’s correction showed that the %FM assessed at the sub-total level as well as in the arms and in the non-impaired leg, was significantly higher in athletes in the VS1 functional class versus able-bodied SV athletes by about 9% (P = 0.016), 15% (P < 0.001) and 8% (P = 0.009), respectively (Fig. 3A). No statistically significant differences were found between athletes in the VS1 functional class and athletes in the VS2 functional class, nor between athletes in the VS2 functional class and able-bodied SV athletes (P > 0.05 for all).

Figure 3 Body composition (A) and performance in the sit-ups test (B) and the modified plank test (C) assessed in the three functional groups.

VS1, which includes athletes with an impairment that more significantly affects the core functions in sitting volley; VS2, which includes athletes with an impairment that more significantly affects the core functions in sitting volley; AB, able-bodied sitting volley athletes; *, P < 0.05; **, P < 0.01; ***, P < 0.001.

As regards the performance assessed in the administered field tests, statistically significant differences between the three sub-groups were found in the sit-ups test and in the modified plank test only. The post hoc analysis with Bonferroni’s correction highlighted that the performance in both the sit-ups test and in the modified plank test was significantly lower in athletes in the VS1 functional class in comparison with able-bodied SV athletes by about 32% (P = 0.004) and 55% (P = 0.049), respectively (Figs. 3B and 3C). No statistically significant differences were found between athletes in the VS1 functional class and athletes in the VS2 functional class, nor between athletes in the VS2 functional class and able-bodied SV athletes (P > 0.05 for all).

On the other hand, no statistically significant between-group differences were found in the seated chest pass test, in the handgrip strength test (executed both with the dominant and the non-dominant hands) and in all four sport-specific field tests (Table 1).

Correlation analysis

After accounting for the assigned group (i.e., VS1 functional group, VS2 functional group and able-bodied SV athletes), no sport-specific field test showed a statistically significant relationship with the considered general characteristics (i.e., age and weekly amount of training) and all the anthropometric variables, with the exception of the hand span. In fact, negative and statistically significant associations were found between the 5m backward sprint test, the modified agility T-test and the speed and endurance test and the hand span (Table 2).

As reported in Table 2, partial correlation analysis also showed positive and statistically significant associations between all four sport-specific field tests and the sub-total %FM and the %FM in the trunk region. Similarly, positive and statistically significant associations were found between all the sport-specific field tests, with the exception of the 5m forward sprint test, and the %FM in the arms region as well as between the modified agility T-test and the speed and endurance test and the %FM in the non-impaired leg (Table 2).

As far as the performance in the physical fitness tests assessing the upper body strength is concerned, negative and statistically significant relationships were found between the 5m forward sprint test and the sit-ups test and between both the 5m forward sprint test and the 5m backward sprint test and the modified plank test (Table 2). Furthermore, negative and statistically significant associations were observed between both the modified agility T-test and the speed endurance test and all the strength field tests with the exception of the modified plank test (Table 2).

The results of the correlation analysis conducted in the sub-groups of athletes, with unilateral lower limb amputation only (n = 17), are represented in Fig. 4. The mean values (± standard deviation) of the impaired leg length and the %FM in the impaired leg region were 61.6 cm (±16.6) and 24.6% (±6.0), respectively. As shown in Fig. 4 (Panels A–H), no statistically significant associations were found between the impaired leg length and the 5m Forward Sprint Test and the 5m backward Sprint Test, while negative and statistically significant correlations were found between the impaired leg length and both the modified agility T-test and the speed and endurance test (Figs. 4C and 4D). Positive and statistically significant associations were also found between the %FM in the impaired leg region and all four of the considered field tests assessing the sport-specific sprint performance (Figs. 4E–4H).

Figure 4 Bivariate correlation analysis conducted in the sub-groups of athletes with unilateral lower limb amputation and classified as VS1 (n D 17).

Scatter plots showed the association between the impaired leg length and the performance in the 5m forward sprint test (A), 5m backward sprint test (B), modified agility T-test (C) and speed and endurance test (D) and the association between the %FM in the impaired leg region and the performance in the 5m forward sprint test (E), 5m backward sprint test (F), modified agility T-test (G) and speed and endurance test (H) r, Pearson’s product-moment correlation coefficient; P, P-value; %FM, DXA-measured percentage of fat mass.

Discussion

Investigating the anthropometric, physical fitness and sport-specific performance of SV athletes across functional class groups is of great importance in their classification and training. Today little research has been conducted on SV (Petrigna et al., 2022) and this is the first study which took into consideration some important physical aspects of SV athletes and their interrelations with sport performance, e.g., the length of the hand and the lower limb segments and body composition.

The aim of this study was twofold: first, to examine the differences in anthropometry, physical fitness and sport-specific sprint performance in SV athletes with respect to their assigned groups (i.e., VS1 functional class, VS2 functional class and able-bodied SV athletes); second, to explore the relationship between the anthropometric and physical fitness characteristics of athletes and their sport-specific sprint performance.

In summary, the results demonstrated the following points:

• Athletes in the three groups had similar body dimensions, while they differ in the %FM that is higher in athletes of the VS1 functional class (i.e., athletes with unilateral lower limb amputation) versus athletes of the VS2 functional class and able-bodied SV athletes.

• No differences were found in the upper limbs strength and in the sport-specific sprint performance across athletes in the three functional class’ groups.

• Greater hand span, greater length of the impaired lower leg (in athletes with unilateral lower limb amputation only), lower %FM at both the sub-total and regional level and higher level of strength in the upper body are all associated with better performances in the four considered sport-specific sprint tests.

Difference in physique and performance across the three groups

When comparing physique and performance of athletes across the three groups (i.e., VS1 functional class, VS2 functional class and able-bodied SV athletes), the results of the present study showed that athletes had a similar mean age and mean hours of weekly training. Considering that all these variables could have an impact on both physique and performance, this result suggests that the three groups were homogeneous from this point of view and, accordingly, comparable. Similarly, no statistically significant differences were found between the three groups in all the considered anthropometric characteristics (Table 1), thereby suggesting that athletes in the three groups were homogeneous with regard to physical dimensions.

As far as body composition is concerned, the results of the present study showed that athletes in the VS1 functional class (i.e., athletes with unilateral lower limb amputation) have higher levels of %FM at both the sub-total and regional levels (i.e., in the arms and in the non-impaired leg) in comparison with able-bodied SV athletes. This result was in line with previous findings (Sherk, Bemben & Bemben, 2010; Cavedon, Zancanaro & Milanese, 2020) and confirmed that people with lower limb amputation undergo a systemic and regional increase in body adiposity. This result underlies the need for nutritionists, clinicians, medical sports doctors and physical conditioners to consider that the type of physical impairment (in this case, the amputation of a lower limb) has an impact on body composition. Accordingly, from a practical perspective, this would for example imply that training programs and nutritional interventions aimed at improving body composition in SV athletes should be distinguished by functional class and specific for the type of the disability.

As regards the performance in the battery of field tests assessing the upper body strength, the performance assessed in the two field tests evaluating the strength of the trunk musculature (i.e., the sit-ups test and the modified plank test) was significantly lower in athletes in the VS1 functional class in comparison with athletes in the VS2 functional class and able-bodied SV athletes. We assume that this result is due to the fact that in the two above-mentioned tests the impairment may have had an impact on the execution of the test. More specifically, athletes with unilateral lower limb amputation would have had a disadvantage in the execution of both tests due to the fact that they only put one foot on the ground thereby reducing the base of support. Accordingly, we think that these two tests were not adequate to assess the strength of the trunk musculature independently from the type of impairment. Future research is therefore needed to create field tests that are more suitable for the assessment of trunk strength (in particular the core musculature) considering that in SV, as well as in most Paralympic sports, athletes have different types and degrees of severity of their impairments.

On the other hand, the performances registered in the Seated Chest Pass Test and in the handgrip strength test executed with both the dominant and the non-dominant hands were similar across the three groups (Table 1), indicating that the three groups had similar strength levels in the upper body. This result was expected because no athlete had an impairment that affected the upper body segments.

Another interesting result of this study was that the performances in all four sport-specific field tests were similar (P > 0.005; Table 1) in the three groups (i.e., VS1 functional class, VS2 functional class and able-bodied SV athletes). In other words, the results suggest that the assigned functional class did not seem to affect the proficiency in the sprint abilities typical of SV. Considering that the three groups were similar for age, weekly amount of training and anthropometric characteristics (Table 1), it is suggested that the severity of impairment in itself could not be associated with performance in the sprint abilities typical of SV. Within each Paralympic sport, athletes should be divided into classes according to the extent of activity limitation caused by their impairment and by minimizing the impact of impairment on the outcome of competition (Tweedy & Vanlandewijck, 2011). The most important guiding principle for setting the number of classes should be that within any given class athletes should not succeed simply because their impairments are less severe than those of their competitors (Tweedy & Bourke, 2009). The absence of statistically significant differences in the considered sport-specific field tests between athletes in the VS1 functional class, VS2 functional class and able-bodied SV athletes, may suggest that the current classification system adopted in SV could not entirely match the actual functional potential of athletes in terms of sport-specific sprint abilities. Taken together these results seem not to confirm the current classification system in SV, i.e., division in two classes: athletes with a disability (i.e., the VS1 functional class) and athletes with a minimal disability (i.e., athletes in the VS2 functional class). These results were in line with previous studies investigating the validity of the current classification system adopted in SV in terms of game efficiency (Marszałek, Molik & Gomez, 2018) and further underlined the need for future research with a larger sample size evaluating the criterion used to divide athletes into the VS1 and VS2 functional classes.

Relationship between physique, strength and sport-specific sprint performance

After controlling for the assigned group, the results revealed that the considered general characteristics of athletes (i.e., age and amount of training) were not associated with their performance in the four sport-specific field tests (i.e., 5m forward sprint test, 5m backward sprint test, modified agility T-test and speed and endurance test). Similarly, the hand span was the only anthropometric variable associated with the performance in three out of four sport-specific field tests. However, previous findings (Marszalek et al., 2015) reported a negative and statistically significant association between the range of reach (i.e., the arm span) and the time to complete the 5m forward sprint test, the T-test and the speed and endurance test. One explanation of this conflicting finding could be due to the differences in the way used to take the anthropometric measurement (i.e., standing as in the case of the study of Marszalek et al. (2015) and sitting as it is in the case of the present study) as well as to the heterogeneity of the study sample. In fact, the sample size of the above-reported study comprised both males (n = 12) and females (n = 8) whose type of disability was not known.

The statistically significant negative association between the hand span and the performance in three out of four sport-specific sprint tests (Table 2) suggests that athletes with greater hand span values, were those who took less time to complete the sprint tests. This result can be explained by the fact that in SV when athletes move in the field the hands act as a support base and are used in the actions of support and propulsion of the body in different directions. It is intuitive to imagine how, from a biomechanical point of view, in SV a greater hand span could represent a more efficient lever system. From a practical perspective, bearing in mind that all disabilities in this study group were in the lower part of their body, this result suggests that impairments, like for example a total or partial hand amputation, could have a negative impact on the sprint abilities typical of SV. Considering that the Paralympic systems of classification aim at promoting participation in sport by people with disabilities at the most appropriate level of rivalry (Doyle et al., 2004; Tweedy & Vanlandewijck, 2011), the question to be raised is what is the impact of a hand impairment on the outcome of SV performance.

Another intriguing result of this study concerning classification regards the negative and statistically significant association we found in the sub-group of athletes with unilateral lower limb amputation between the length of their impaired leg and the time to complete the two sprint tests with changes of directions (i.e., the modified agility T-test and the speed and endurance test; Figs. 4C and 4D). This result suggests that, in athletes with amputation, the level of amputation could an impact on a key performance outcome, where athletes with an above-knee amputation seem to be more disadvantaged in comparison with athletes with below-knee amputation. It is important to underline that both athletes with an amputation above the knee and athletes with an amputation below the knee all compete within the same functional class (i.e., the VS1 functional class). Accordingly, it would be interesting to further investigate the impact of the level of amputation on the outcome of other SV abilities like for example in the execution of some technical fundamentals of the game (e.g., the serve). This would help to understand whether within the VS1 Class, the impact of impairment is minimized on the outcome of competition. When dealing with Paralympic athletes, it is always important to bear in mind that winning or losing a competition should always be dependent on training, talent, motivation, and skill, rather than on belonging to a favoured or disadvantaged Class (Tweedy & Vanlandewijck, 2011). Taken together these results open the way for future research with a larger sample size aimed at considering further criterion which could be used in SV to attribute the functional classes to athletes (e.g., consideration of above or below the knee amputation, impairments affecting the hand).

An important finding of this study inherent to body composition is that, in addition to the fact that fat accumulation has negative consequences from a health perspective (Anderson et al., 2013), in SV, independently from the functional class, higher %FM at the sub-total and at the regional level seems to be associated with worse performance in the sport-specific field tests, in particular in those requiring changes of directions (Table 2). Interestingly enough, in the only group of athletes with an amputation (VS1 functional class) the %FM assessed in the impaired leg has a positive and statistically significant association with all the considered sport-specific field tests (Figs. 4E–4H), suggesting that a greater fat mass accumulation in the impaired leg has a negative impact in the sport-specific sprint ability. These results suggest that, regardless of the severity of the impairment, body composition has an impact on the SV sprint performance. Accordingly, based on these results, physical conditioners, coaches and nutritionists are encouraged to develop training programs as well as nutritional strategies aimed at improving body composition in SV athletes. In particular, training programs should include specific exercises that target the musculature of the lower limbs, including the impaired leg in athletes with amputation.

When considering the association between the performance in the upper body strength field tests and the performance in the four sport-specific sprint tests, partial correlation analysis showed that, after controlling for the assigned group, negative associations were found between the performance in both the sit-up tests and the modified plank test and the performance in all sport-specific sprint tests (Table 2). Specifically, better performances in the sit-ups test and in the modified plank test were associated with better performances in the sport-specific sprint tests. This result suggests that in each group athletes should be trained with exercises targeting the musculature of the trunk in order to improve their sprint performance.

A finding of this study was that negative and statistically significant associations were found between the field tests adopted to assess the bilateral upper arm strength and the performance in the two sport-specific sprint tests with changes of direction (i.e., the modified agility T-test and the Speed and endurance test; Table 2). Accordingly, athletes with higher strength levels in their upper limbs were the ones who were faster in sprinting in different directions. This result should encourage physical conditioners and coaches to include exercises to strengthen the upper body musculature in their training programs in order to improve sprint performance with direction changes. It is surprising that the upper body strength seems to be relevant only in the sprints with changes of direction and that no association was found between the performance in both the seated chest pass test and the handgrip strength test and the performance in the two sport-specific sprint tests based on straight sprinting (i.e., the 5m forward sprint test and the 5m backward sprint test). It is reasonably to argue that there may be other factors associated with performance in straight sprinting in SV, that is for example the strength of the lower part of the body. In this study we did not measure the strength of the lower body but, based on the results of the present study, in the future it would also be interesting to assess the association between the strength of the lower body and the sport-specific sprint performance.

This study has some limitations to be mentioned. From a statistical point of view, a first limitation is the moderate sample size (n = 35). In fact, the sample size was only adequate to detect a medium/large effect size (η2 = 0.23) with an acceptable power of 80%. Accordingly, studies with larger sample sizes are required to investigate smaller effect sizes. The second limitation is the type of field tests adopted to evaluate the trunk strength whose performance would be affected by the type of impairment. Considering the variety of impairments typical within the SV community, in order to reduce the impact of the impairment on the outcome of the test performance, future research is needed to evaluate the trunk strength using for example an isokinetic dynamometer according to previous literature on amputee soccer players (Aytar et al., 2012). A third limitation of this study was that we did not include tests to evaluate the strength of the lower limbs (both impaired and non-impaired). Considering the results of this study (i.e., the association between the %FM in the lower limb and the sport-specific sprint performance), in a future study it would be interesting to evaluate the impact of the strength of the lower limb on the performance in sport-specific sprints.

This study has also some strengths to underline. First, we tried to mitigate variability as much as possible by recruiting athletes of the same gender (i.e., males) and all with a lower limb physical impairment. Moreover, it is important to underline that in the VS1 group all athletes had the same type of physical impairment (i.e., a lower limb amputation). Second, to the best of our knowledge, this is the first study investigating body composition using DXA (i.e., a reference method) in SV athletes, thereby allowing insight into peculiar regional compositional characteristics of this athletic population.

Conclusions

In conclusion, the results of the present study have two important practical implications, one regarding the design of training programs in SV and the other concerning the validity of the current system of classification adopted in SV. Based on the above results, professionals dealing with SV athletes should consider strategies aimed at improving body composition specific for athletes in the VS1 functional class with a lower limb amputation and, regardless of the functional class, they are encouraged to include specific exercises aimed at improving body composition in the lower limbs and the strength of the trunk and upper limbs in their training programs. From a classification perspective, these results do not confirm the validity of the current system of classification of athletes adopted in SV and suggest the need for a thorough assessment of some of the points raised in this study. Ensuring fair and equitable competitions between athletes with different impairment types and severities is essential to promote the practice of adapted sports, wider inclusion and the full ethical principles of sport. This is even more important in countries where initiatives specific for SV players are already present, the main aim is that of recreation and socialisation but this does not detract from the fact that competition is available and should be regulated to create a fair and unbiased structure for all those who participate. In fact, participation in SV by athletes with an impairment and able-bodied athletes should be encouraged and facilitated promoting an appropriate evidence-based classification of athletes on the basis of their functional and performance abilities.

Supplemental Information

Data S1 Raw data

Click here for additional data file.

The authors would like to thank all athletes who participated in the study, their coaches and the FIPAV (Federazione Italiana Pallavolo) for their kind cooperation.

Additional Information and Declarations

Competing Interests

Author Contributions

Human Ethics

Data Availability

The authors declare there are no competing interests.

Valentina Cavedon conceived and designed the experiments, performed the experiments, analyzed the data, prepared figures and/or tables, authored or reviewed drafts of the article, and approved the final draft.

Chiara Brugnoli conceived and designed the experiments, performed the experiments, authored or reviewed drafts of the article, and approved the final draft.

Marco Sandri analyzed the data, prepared figures and/or tables, authored or reviewed drafts of the article, and approved the final draft.

Luciano Bertinato performed the experiments, authored or reviewed drafts of the article, and approved the final draft.

Lorenzo Giacobbi performed the experiments, authored or reviewed drafts of the article, and approved the final draft.

Filip Bolčević performed the experiments, authored or reviewed drafts of the article, and approved the final draft.

Carlo Zancanaro conceived and designed the experiments, authored or reviewed drafts of the article, and approved the final draft.

Chiara Milanese conceived and designed the experiments, authored or reviewed drafts of the article, and approved the final draft.

The following information was supplied relating to ethical approvals (i.e., approving body and any reference numbers):

The study protocol was approved by the Institutional Review Board of the Verona University (Ethical Application Ref: Protocol number: 18198, 05/04/2013).

The following information was supplied regarding data availability:

The raw measurements are available in the Supplementary File.

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
