# Peer review of "Physique and performance in male sitting volleyball players: implications for classification and training"

_PeerJ, doi:10.7717/peerj.14013_

## Round 0.1 · original submission · Major Revisions

Thank you for submitting the manuscript to PeerJ. It has been reviewed by experts in the field and we request that you make major revisions before it is processed further.

We look forward to hearing from you soon.

Best wishes,

Dr Shibili Nuhmani

·

Basic reporting

The article has been written in English and has used clear, unambiguous, technically correct text. The article has conformed to professional standards of courtesy and expression. The article has included sufficient introduction and background to demonstrate how the work fits into the broader field of knowledge. Relevant prior literature has been appropriately referenced. The submission has been self-contained, represented an appropriate ‘unit of the publication’, and included all results pertinent to the hypothesis.

Experimental design

The submission has clearly defined the research question, which must be relevant and meaningful. The knowledge gap being investigated has been identified, and statements have been made as to how the study contributes to filling that gap.

Validity of the findings

no comment

Reviewer 2 ·

Basic reporting

This study aimed to assess if anthropometry, physical fitness and sport specific sprint performance varied between groups of sitting volleyball athletes in order to verify the validity of the System of Classification. The second objective was to investigate how the anthropometric and physical fitness characteristics of the athletes relate to the sport-specific sprint performance. The topic is interesting and the community needs papers like this. It is well written and presented, all the sections are referenced. The methodology adopted is complete and accurate. I have only some minor comments:

Considering the length of the introduction, I suggest to make the first paragraph shorter bringing the attention of the reader to the topic of this research. I further suggest to update the revision of the literature, from January to today studies have been published on sitting volleyball. I just read a review on this topic that I suggest to include.
Line 332: please remove “was found”

Experimental design

The experimental design is well written, structured and presented

Validity of the findings

The findings are well presented and well supported by a good statistical analysis

Reviewer 3 ·

Basic reporting

The authors addressed an interesting study. They investigated anthropometry, physical fitness and sport specific sprint performance across the three groups of functional classes in SV finding differences on the abovementioned characteristics. Moreover, they would like to investigate how the anthropometric and physical fitness characteristics of athletes relate to their sport-specific sprint performance.

The introduction provides a sufficient and suitable background and authors have clearly defined aim and hypothesis.

My major doubt for this study concerns the small sample considered for the study. The authors recruited 35 participants including 17 athletes with a disability [VS1], and only 9 for each athletes with a minimal disability [VS2] group and able-bodied SV athletes [AB] group. In my opinion, I do not think these numbers are sufficient to question the validity of the current system of Classification of athletes adopted in sitting volley, as reported by the authors. Authors should be more cautious in hypothesizing this and even more cautiously in interpreting the data.
In that regard, could they provide a post-hoc power analysis of the sample?

Experimental design

Authors should report the type of study design.

The procedure should be replicable. In this regard, I have the following comments to make:

The type of sampling and how the participants were recruited should be defined.

The setting of the procedure is not clear. The authors state that "Testing took place on the same day, in the late morning/early afternoon, after a 3-4 h fast." (line 160). However, they report to have measured body composition using a DXA total body scanner in their laboratory (line 196-200) and physical tests in a indoor gym (line 213).

Could the authors explain why between the three hand grip strength trials they assigned a rest of 2-5 seconds? Since each trial address to measure a maximal strength, could they provide literature to support the choice of this rest time?

Again, why authors choose 45-60 second rest between sit-ups trials? it's enough?

Validity of the findings

As for statistical analysis, I suggest that the Shapiro–Wilk test is more appropriate considering the small sample size.

Please provide the post-hoc sample size power achieved.

In confirmation of my greatest doubt concerning the small sample size, the authors performed the correlation analysis between between general characteristics, anthropometry, physical fitness and sport-specific sprint performance only for the VS1 group.

These results do not allow to support what the authors hypothesized, although this hypothesis could be valid if the sample were increased and the results confirmed (extending the correlation analyzes also for the other functional classes).

---

## Round 0.2 · accepted · Accept

Dear Authors,

Your manuscript has been improved. It is now worthy of publication in PeerJ.

Congratulations!